# Management of Inflammatory Bowel Disease with History of Cancer

**DOI:** 10.3390/cancers17213475

**Published:** 2025-10-29

**Authors:** Vito Annese, Marzio Parisi, Sofia Cinque, Alessandro Cappellini, Paolo Biamonte, Giuseppe Dell’Anna, Sabrina Gloria Giulia Testoni, Maria Laura Annunziata

**Affiliations:** 1Unit of Gastroenterology, IRCCS San Donato Policlinic, San Donato Milanese, 20097 Milan, Italy; sof.cinque@studenti.unina.it (S.C.); paolo.biamonte@grupposandonato.it (P.B.); giuseppe.dellanna@grupposandonato.it (G.D.); sabrina.testoni@grupposandonato.it (S.G.G.T.); marialaura.annunziata@grupposandonato.it (M.L.A.); 2Department of Gastroenterology and Endoscopy, Vita-Salute University San Raffaele, 20132 Milan, Italy; 3Department of Internal Medicine and Medical Therapy, University of Pavia, 27100 Pavia, Italy; ignaziomarzio.parisi01@universitadipavia.it (M.P.); alessandro.cappellini01@universitadipavia.it (A.C.); 4Department of Clinical Medicine and Surgery, Gastroenterology Unit, University Federico II, 80131 Naples, Italy

**Keywords:** inflammatory bowel disease, Crohn’s disease, ulcerative colitis, cancer risk, melanoma, lymphoma, advanced therapy, biological therapy, management

## Abstract

**Simple Summary:**

Crohn’s disease and ulcerative colitis, the main inflammatory bowel diseases (IBDs), have experienced rising prevalence worldwide, and their management is still challenging despite many therapeutic opportunities. Patients with IBDs have the same background risk of experiencing cancer but are also exposed to increased risk for some of them, i.e., colorectal cancer. A previous or current history of cancer is a dilemma for the IBD specialist since these patients are never included in a randomized controlled trial. Therefore, their management is based on multidisciplinary evaluation, expert consensus, and real-world experience. This review aims to clarify this situation and offer some clues on patient management.

**Abstract:**

**Background/Objectives**: Because of the chronic course of the disease, clinicians managing IBD frequently encounter patients with a prior or newly diagnosed cancer. This can be related to the specific background cancer risk in that subject, aging, familial/genetic factors, or ongoing chronic inflammation. However, a potential influence of some therapeutic agents should also be considered. This setting, in the absence of controlled trials and few open series reports available, raises issues such as correct screening, prevention, and surveillance, but also eventual modification or adaptation of the medical management. **Methods and Results**: Few consensus guidelines and studies are available on the management of IBD patients with a history of cancer, and therefore, we aim to review the recommendations of the current guidelines and the evidence reported in the most recent real-world cohorts. **Conclusions:** This review will offer (a) an understanding of the background of cancer risk in IBD patients; (b) analysis and discussion of the risk of cancer related to IBD therapy; and, finally, (c) some clues for the management of IBD in patients with a previous or current history of cancer.

## 1. Introduction

Due to the chronic course of inflammatory bowel disease (IBD) and its often early onset, clinicians frequently encounter patients with a history of, or newly diagnosed, malignancy. This increased risk may be influenced by various factors, including age-related cancer susceptibility, genetic or familial predisposition, and the persistent inflammatory environment characteristic of IBD. Additionally, certain therapeutic interventions may carry potential carcinogenic effects. In this context—where randomized controlled trials are limited and observational data remain scarce—careful consideration is required for cancer screening, preventive strategies, surveillance protocols, and potential modification of treatment regimens. The recommendations and discussion provided in this review are primarily informed by the European Crohn’s and Colitis Organization (ECCO) consensus statements, initially published by Annese et al. in 2015 [1] and subsequently updated by Gordon et al. in 2023 [2]. In addition, two independent senior authors (VA and MLA) performed a comprehensive search of the PubMed, Embase, and Scopus databases up until 30 September 2025 to identify articles published in the English language on this topic. Search terms such as “cancer”, “malignancies”, history of cancer”, “history of malignancies” in combination with “Crohn’s disease”, “ulcerative colitis”, and “inflammatory bowel disease” were used.

## 2. Background Risk of Cancer in IBD

A recent meta-analysis encompassing 20 population-based studies [3], along with a contemporary Scandinavian cohort [4], estimated that individuals with IBD have a 1.4–1.7-fold increased risk of colorectal cancer (CRC). Although CRC incidence has shown a gradual decline over time [5,6,7,8], it remains elevated, potentially reflecting improvements in endoscopic surveillance and more effective inflammation control [7,9]. In ulcerative colitis (UC), the standardized incidence ratio (SIR) is particularly high in patients diagnosed at a younger age (SIR = 8.6) and in those with extensive colonic involvement (SIR = 4.8). Additional risk factors include a family history of CRC and the coexistence of primary sclerosing cholangitis (PSC), especially among UC patients [10].

For Crohn’s disease (CD), the estimated SIR for CRC is 2.4, with disease location strongly influencing cancer risk [11]. Although the absolute risk of small bowel cancer remains relatively low and varies across studies, its incidence in CD is estimated to be at least ten times higher than in the general population [3,5].

Notably, IBD patients tend to develop CRC at a younger age than individuals without IBD [12], and those diagnosed before 50 years exhibit higher mortality and reduced five-year survival [13]. Overall, CRC-related mortality is approximately 50% higher in IBD compared to the general population [4,14].

Although uncommon, CD patients may develop adenocarcinoma arising from longstanding perianal or enterocutaneous fistulas [15]. Similarly, UC patients with ileal pouch-anal anastomosis (IPAA) may rarely experience pouch-related malignancy [16]. Incidence rates of other gastrointestinal cancers appear to be rising faster in IBD than in age- and sex-matched non-IBD populations [8]. A 25-year population-based study from Ontario reported that the average annual percentage change (AAPC) in cancer incidence increased more rapidly in IBD, particularly for liver and bile duct cancers [8]. Beyond 2010, both the incidence and mortality of bile duct (AAPC = 2.3, 95% CI 1.96–2.77) and pancreatic cancers (AAPC = 1.19, 95% CI 1.0–1.4) were elevated in IBD patients [8]. UC patients with concurrent PSC are at especially high risk for cholangiocarcinoma [17].

Beyond the gastrointestinal tract, IBD is associated with higher rates of extraintestinal malignancies compared to the general population [18]. By tumor site, CD has been linked to increased risks of upper gastrointestinal malignancies, lung cancer, bladder cancer, and non-Hodgkin lymphoma, whereas UC shows a higher incidence of hepatobiliary cancers and hematologic malignancies, such as leukemia [11,19,20,21,22,23]. Both UC and CD are associated with higher rates of melanoma and non-melanoma skin cancers (NMSCs), with advancing age further elevating risk [19,24,25,26,27,28]. Conversely, the incidence of breast and prostate cancers does not appear to differ significantly from the general population [29]. Importantly, current evidence does not demonstrate a definitive link between advanced IBD therapies and the development of extraintestinal cancers [30] (see Table 1).

### Management of Background Risk of Cancer

Surveillance colonoscopy programs play a critical role in reducing morbidity and mortality from CRC by facilitating early detection, improving prognosis, and allowing for the removal of dysplastic lesions, thereby decreasing CRC incidence. A Cochrane review of observational studies including over 7000 IBD patients demonstrated lower cancer detection rates in patients undergoing surveillance compared to those not screened (OR = 0.58, 95% CI 0.42–0.80) [31]. In addition, CRCs in the surveillance group were diagnosed at earlier stages (OR = 5.4, 95% CI 1.51–19.30) and were associated with reduced CRC-related mortality (OR = 0.36, 95% CI 0.19–0.69) [31]. These findings have been confirmed in recent case series [32,33,34].

The estimated benefit, expressed as life-years gained, is substantially higher in IBD patients compared to average-risk populations, as IBD-related CRC tends to occur at younger ages. Modeling studies suggest that each screening colonoscopy can yield 1.2–5 life-years in UC patients, compared to only 1.2–4 months in average-risk populations [35,36]. Recent declines in CRC incidence likely reflect both effective surveillance and improved disease control. Because disease duration is a key determinant of CRC risk, initial screening colonoscopy is generally recommended after approximately eight years from disease onset [37]. This first colonoscopy also provides an opportunity to reassess disease extent, which influences subsequent CRC risk.

The frequency of follow-up colonoscopies should be tailored to the individual’s risk of dysplasia progression, although the natural history of progression in IBD remains incompletely understood. Intervals should be adjusted according to patient-specific risk factors and prior endoscopic findings, with disease extent defined as the most extensive histologically confirmed inflammation across all previous colonoscopies. Risk stratification and corresponding surveillance intervals have been outlined by the British Society of Gastroenterology and are endorsed by ECCO [2,38,39,40,41] (Table 2).

Recent advances in endoscopic technology, bowel preparation, and diagnostic techniques have markedly improved surveillance quality. Nonetheless, post-colonoscopy CRC rates remain significantly higher in IBD than in the general population (pooled 3-year rate: 30.8% vs. 6.8%) [42]. Contributing factors may include rapid cancer progression, suboptimal surveillance intervals, and procedural challenges such as poor bowel preparation [41]. High-definition (HD) endoscopes improve visualization and may enhance dysplasia detection [43,44].

Dye-based chromoendoscopy (DCE) using methylene blue or indigo carmine [45], as well as virtual chromoendoscopy (VCE) techniques (e.g., i-SCAN, NBI, and BLI) combined with targeted biopsies, substantially increase the detection of dysplastic lesions [46,47,48] and are recommended for IBD surveillance. A recent network meta-analysis suggested that HD-DCE may outperform HD white-light endoscopy, though the certainty of evidence is low, with no significant difference observed between HD-DCE and HD-VCE [49]. These approaches enhance identification of both polypoid and flat lesions, reducing the need for random biopsies of normal-appearing mucosa. Optimal bowel preparation is critical, and longer withdrawal times have been associated with higher adenoma detection rates in non-IBD populations [42].

The latest BSG guidelines [41] also emphasize the importance of organized surveillance programs, including personalized reminders and risk-based surveillance intervals. To support risk stratification, an online multivariate risk calculator based on large North American and European datasets has been provided (https://ibd-dysplasia-calculator.bmrc.ox.ac.uk, (accessed on 30 September 2025)) [50].

Artificial intelligence (AI) tools using computer-aided detection (CADe) and diagnosis (CADx) have not yet been validated for dysplasia detection in IBD and are not currently recommended, though ongoing developments are expected [41].

For extraintestinal malignancies, no specific screening guidelines exist beyond general preventive measures such as smoking cessation, annual dermatologic evaluations, and rigorous photoprotection. These interventions are particularly critical for patients receiving thiopurines or biologic therapies [51,52,53,54,55,56].

## 3. Therapy for IBD and Risk of Cancer

Malignancies directly attributable to immunosuppressive or so-called “advanced” therapies (biologics and small molecules) account for only a minority of incident cancers observed in patients with inflammatory bowel disease (IBD) (see Table 3).

### 3.1. Thiopurines

Thiopurines may promote carcinogenesis through several mechanisms, including induction of DNA mutations, impairment of tumor immune surveillance, reduction in immune cell quantity and function, and stimulation of proliferation in cells with microsatellite instability [57]. Recent meta-analyses reported a significantly increased SIR for lymphoma in thiopurine-exposed patients (SIR 5.7; 95% CI 3.2–10.1), whereas no elevated lymphoma risk was observed in former users or patients never exposed to these drugs [58,59]. Absolute lymphoma risk was approximately two to three times higher in men than women, independent of age or duration of exposure, with the highest absolute risks in patients older than 50 years (2.6 per 1000 patient-years) and males under 30 (1–2 per 1000 patient-years). Analyses of treatment duration suggest that lymphoma risk attributable to thiopurines does not substantially increase beyond the first year of therapy.

Thiopurines have also been linked to long-term risk of acute myeloid leukemia and severe myelodysplastic syndromes [60]. Hepatosplenic T-cell lymphoma (HSTCL), a rare but aggressive complication, occurs almost exclusively in males under 35 treated with thiopurines—either alone (1 per 10,000 patient-years) or in combination with anti-TNF agents (3 per 10,000 patient-years)—for more than two years [58]. Limiting combination therapy duration to two years, when clinically feasible, may mitigate this risk [61].

Multiple recent studies and a meta-analysis have associated thiopurine therapy with increased NMSC risk in IBD, with a pooled adjusted hazard ratio of 2.3 [62]. Interestingly, the risk persisted after thiopurine discontinuation in the CESAME study [63], but this finding was not replicated [64].

The widespread introduction of advanced therapies has prompted renewed attention to thiopurine-associated cancer risk. A Swedish nationwide analysis of approximately 64,000 UC patients with a median follow-up of 8 years reported that 11,916 received thiopurines, while 3452 received combination therapy with anti-TNF agents [65]. Compared to the general population, 2.7 excess cancer cases per 1000 person-years (HR = 1.12, 95% CI 1.09–1.16) occurred with thiopurine monotherapy, and combination therapy (HR = 1.44, 95% CI 1.19–1.75). Excess cancers included colorectal, hepatobiliary, lymphoid, and basal cell skin cancers. Similarly, a Danish registry following 43,419 IBD patients over a median of 8.2 years reported increased cancer risk with thiopurine monotherapy (adjusted hazard ratio [aHR] 1.36, 95% CI 1.17–1.57) and combination therapy with anti-TNF agents (aHR 2.49, 95% CI 1.64–3.78) [66]. Specific risks were noted for melanoma, lymphoid, urinary tract, female genital, breast, and NMSC cancers. Longer exposure (>5 years) increased hazard ratios, but after drug discontinuation, risk returned to baseline.

### 3.2. Methotrexate and Cyclosporine

Robust data on malignancy risk with methotrexate or cyclosporine in IBD remain limited. Evidence from rheumatologic cohorts does not indicate increased incidence of solid tumors or hematologic malignancies with methotrexate. Although calcineurin inhibitors are associated with higher cancer risk in transplant recipients, this effect is dose- and duration-dependent and generally not relevant at the doses used for IBD.

### 3.3. Anti-TNFα Agents

Tumor necrosis factor-alpha (TNF-α) inhibition was initially hypothesized to increase cancer risk by impairing tumor immune surveillance. The evaluation of multiple studies is complicated by prior or concurrent thiopurine exposure and inadequate statistical power. Recent large nationwide Danish studies, in line with meta-analyses, have not demonstrated an overall increased cancer risk with infliximab or adalimumab [67,68,69,70], including in patients over 60 years [30]. The effect of anti-TNF therapy on thiopurine-associated lymphoma risk remains conflicting, except for HSTCL [71,72,73].

A Swedish nationwide study documented modest excess cancer cases with anti-TNF therapy (2.7 per 1000 person-years; HR = 1.41, 95% CI 1.24–1.62), primarily due to colorectal and hepatobiliary cancers, likely reflecting more aggressive disease rather than therapy [65].

Melanoma incidence is rising in developed countries. In a large nested case–control study, TNF-α antagonists were modestly associated with melanoma risk (OR 1.9; 95% CI 1.1–3.3) [25], although Danish cohort data did not confirm this association [69]. Systematic reviews in IBD, rheumatoid arthritis, and psoriasis populations similarly do not support a significant link between anti-TNF therapy and melanoma [54]. NMSC risk with anti-TNF therapy is less clear; a systematic review of nearly 300,000 IBD patients reported a 17.8% NMSC rate, consistent with the general population [66], whereas Swedish registry data indicate an elevated risk of basal cell carcinoma (HR = 1.62; 95% CI 1.21–2.03) [65].

### 3.4. Anti-Integrins

Vedolizumab is a humanized IgG1 monoclonal antibody that selectively inhibits lymphocyte migration to the gut by blocking α4β7 integrin–MAdCAM-1 interactions on intestinal vascular endothelium, thereby preventing activated T-cell migration to inflamed gut tissue. It has demonstrated efficacy in moderate-to-severe UC, CD, and pouchitis [74]. A meta-analysis of 88 observational studies including over 25,000 IBD patients found a pooled malignancy rate of 0.3%, not differing from population expectations (*p* < 0.0001, Egger’s test) [75]. Swedish registry data corroborate the absence of increased cancer risk in vedolizumab-treated patients [65].

### 3.5. Anti-IL 12/23

Ustekinumab, a humanized monoclonal antibody targeting IL-12 and IL-23, has shown long-term safety and efficacy in IBD. Pooled data from phase III trials with follow-up up to 5 years in CD and 4 years in UC reported malignancy rates similar to the SEER 2019 database of the National Cancer Institute, including NMSC (ustekinumab 0.32% vs. placebo 0.39% per 100 patient-years), with no lymphoma cases [76]. Real-world data from the POLAR psoriasis registry (>12,000 patients) and Swedish registry also confirmed malignancy rates (excluding NMSC) comparable to population expectations [65,77].

### 3.6. JAK Inhibitors

Janus kinase (JAK) inhibitors are a newer class of immunomodulatory agents used across several immune-mediated inflammatory diseases. Tofacitinib, approved in 2012 for rheumatoid arthritis and more recently for UC, has been studied extensively regarding malignancy risk, but findings remain inconsistent. Concerns regarding tofacitinib’s oncogenic potential have been raised following the publication of the ORAL surveillance study in rheumatoid arthritis [78]. A recent meta-analysis by Bezzio et al. [79] pooled data from 26 studies (including 22 randomized controlled trials) with over 12,000 patients (approximately 2000 with IBD). Among 13 studies reporting malignancies, overall cancer risk did not differ significantly between tofacitinib and placebo or biologics. However, a slightly increased relative risk was observed in tofacitinib-treated patients versus anti-TNF recipients for overall cancers (RR 1.4), melanoma (RR 1.47), and NMSC (RR 1.3). More recently approved JAK inhibitors, such as filgotinib and upadacitinib (the latter for both UC and CD), lack long-term malignancy safety data but have not raised new safety concerns so far [80,81]. A recent systematic review on JAK inhibitors and risk of cancer in IBD concluded [82] that current evidence does not indicate the need for heightened oncologic vigilance beyond what is routinely applied with other therapeutic classes used in IBD treatment. However, longer observation periods and an increased volume of real-world experiences are awaited.

### 3.7. New Advanced Therapies

Sphingosine 1-phosphate (S1P) receptor modulators, such as ozanimod and etrasimod, limit lymphocyte trafficking to inflamed tissues. Phase III trials and open-label extensions have not shown excess malignancies [83,84,85]. Selective IL-23p19 inhibitors (mirikizumab, risankizumab, guselkumab) are approved for UC and/or CD, and pivotal trials have not identified new cancer safety signals; long-term real-world data are awaited [86,87,88].

### 3.8. Management of Cancer Risk Related to Therapy

Three lymphoma subtypes have been described in thiopurine-treated IBD patients:-Post-transplant-like lymphomas: Predominantly in adults over 30 or EBV-seropositive teenagers.-Post-mononucleosis lymphomas: Occur in EBV-seronegative males who subsequently seroconvert.-Hepatosplenic T-cell lymphomas (HSTCL): Mainly men under 35 on thiopurines, alone or with anti-TNF, for >2 years [1,2].

Careful consideration of thiopurine indication and treatment duration is advised in these groups.

Skin cancer risk factors include age, sex, smoking, fair skin, red hair, cumulative sun exposure, childhood sunburns, outdoor occupations, atypical moles, family history, Caucasian race, geographic location, and genetic predisposition. These should be evaluated before initiating immunosuppressive therapy. Alternative treatments to anti-TNF agents and calcineurin inhibitors may be preferable in melanoma survivors or high-risk patients. Thiopurines should be avoided in those with a history of aggressive NMSC [1,2].

Ongoing studies, such as the ECCO-supported I-CARE study, will provide further insight into advanced therapy safety, including cancer risk [89].

## 4. Management of IBD Patients with a History of Cancer

In patients considered cured of malignancy, clinicians must carefully assess the risk of recurrence or metastatic spread. Data from the SEER registry suggest that cancer survivors face a 14% increased likelihood of developing a second malignancy relative to the general population, with childhood cancer survivors experiencing a sixfold lifetime risk of a subsequent malignancy [90].

Managing IBD in patients with a prior or concurrent history of cancer presents considerable challenges, often complicated by uncertainty among oncologists regarding immunosuppressive therapy. Optimal care requires close coordination between gastroenterologists and oncologists, with individualized decisions based on cancer type, stage, prognosis, and the severity of IBD. Patients with a previous history of cancer have approximately a twofold higher risk of developing new or recurrent malignancies compared to those without prior cancer, independent of immunosuppressant exposure. Key clinical questions to address include the following:
How do IBD treatments influence cancer progression or recurrence?How do cancer therapies affect IBD activity?How should IBD therapy be adjusted in patients with previous, current, or recurrent cancer?

### 4.1. Immunosuppressant Therapy and Cancer Recurrence

Most evidence regarding the safety of immunosuppressants originates from observational studies in rheumatology and transplant populations. Thiopurines, in particular, carry a heightened risk of cancer recurrence, especially for melanoma and NMSC, with recurrence rates exceeding 20% and peaking at 54% within two years post-chemotherapy before gradually declining [91]. The risk varies according to cancer type, as demonstrated in transplant recipients [92].

The CESAME cohort showed that, although prior malignancy increases the overall risk of new or recurrent cancer in IBD patients, the use of immunosuppressants did not significantly alter this risk. These findings, however, should be interpreted cautiously due to the relatively small number of cancer survivors in the study [93]. Evidence on methotrexate and cancer recurrence in IBD remains limited [2].

### 4.2. Use of Anti-TNF and Other Biologics in Cancer Survivors

Current data suggest that anti-TNF therapies can be administered safely in IBD patients with a history of cancer, though decisions should be individualized and made in multidisciplinary settings. A meta-analysis including 11,679 cancer survivors, of whom 3707 were treated with anti-TNF agents post-cancer diagnosis, reported no increased risk of cancer recurrence compared to unexposed individuals [94]. Similar findings have been observed in other immune-mediated conditions [95,96], although study heterogeneity warrants cautious interpretation.

Vedolizumab has not been associated with increased incidence of new or recurrent malignancy in small-scale studies [96,97,98]. Likewise, ustekinumab appears to carry a cancer recurrence risk similar to that of patients not receiving immunosuppressants [98]. A meta-analysis of 31 studies encompassing 24,328 patients over 85,784 person-years indicated comparable recurrence rates between patients on anti-TNF agents and those not receiving immunosuppressants, with vedolizumab and ustekinumab showing numerically lower recurrence rates [99].

Isufi et al. conducted a systematic review of 20 studies involving 4376 patients with immune-mediated diseases and prior cancers, eight of which included IBD patients [100]. Across 15,646 patient-years, anti-TNF exposure (4462 patients) was associated with a recurrence RR of 1.09 (95% CI 0.78–1.52). In IBD patients specifically, RR was 1.00 (95% CI 0.43–2.33, *p* = 0.99) compared to conventional or no therapy. Vedolizumab treatment yielded an RR of 0.70 (95% CI 0.07–7.61) for cancer recurrence. Additionally, ten studies evaluated new cancer incidence: anti-TNF therapy RR was 1.02 (95% CI 0.70–1.50), and vedolizumab RR was 0.44 (95% CI 0.33–0.60).

These results are generally reassuring, though further research is warranted. A multicenter retrospective study of 207 IBD patients with a prior history of breast cancer followed for a median of 71 months found no significant difference in cancer recurrence between patients treated with biologics or immunosuppressants and those who were not [101]. Interim analysis of the SAPPHIRE registry [102], a prospective cohort of IBD patients with previous malignancy on immunosuppressive therapy, included 210 patients exposed to antimetabolites (24%), anti-TNF agents (49%), anti-integrins (44%), anti-IL-12/23 (30%), or JAK inhibitors (6%) over a median follow-up of five years. New or recurrent cancers occurred in 15% of patients, with adjusted analysis showing a non-significant numerical increase associated with immunosuppressant use (aHR 1.41; 95% CI 0.69–2.90).

### 4.3. Influence of Cancer Treatments on IBD Course

Evidence regarding the impact of cancer therapies on IBD is limited. Radiation therapy appears generally safe, with no significant increase in IBD flares, hospitalizations, or surgical interventions [103]. Hormonal therapies, however, may approximately double the risk of IBD exacerbations [104]. In patients with active IBD at cancer diagnosis, immunosuppressive effects of cancer therapy may induce or maintain remission even after discontinuation of IBD-specific medications [1].

A recent systematic review and meta-analysis of 1298 IBD patients receiving cancer therapy [105] reported an overall flare rate of 30% (95% CI 23–37%), leading to corticosteroid use in 25%, biologic therapy in 10%, and discontinuation of cancer therapy in 14%. Subgroup analysis revealed significant heterogeneity (*p* < 0.01) across treatment modalities. Radiation therapy alone was associated with a pooled flare rate of 20% (95% CI 9–38%), generally manageable, whereas chemotherapy produced a pooled flare rate of 48% (95% CI 3–97%). Hormone therapy combined with chemotherapy significantly increased flare risk (HR 2.35; 95% CI 1.25–4.42). Immune checkpoint inhibitors (ICIs) present a particular challenge: among 14 studies, the pooled flare rate was 33% (95% CI 25–42%), with gastrointestinal adverse events more frequent in IBD patients than in the general population. Flares were numerically more common in UC than in CD. Baseline disease activity may not fully predict flare risk, as the largest study reported that 75% of patients with inactive IBD remained in remission after chemotherapy [105].

### 4.4. Management of Therapy in Patients with IBD and Cancer

Currently, no definitive guidelines exist for IBD management in patients with prior or active cancer, though multiple reviews [106,107,108,109,110] and the AGA clinical practice update [111] provide guidance. Ongoing registries, including SAPPHIRE [102] and I-CARE [89], are expected to offer additional insights. Until more evidence emerges, management should be individualized, with multidisciplinary input considering IBD activity, cancer type, stage, prior therapies, and patient-specific factors (Table 4).

Second malignancies remain a significant and potentially lethal complication, comprising roughly 18% of incident cancers in the US. Immunosuppressive therapies—including thiopurines, calcineurin inhibitors, anti-TNF agents, and JAK inhibitors—are generally withheld until cancer therapy is completed. In contrast, 5-aminosalicylates, nutritional therapy, and topical corticosteroids (e.g., budesonide) are considered safe for active IBD in this context. Severe flares unresponsive to these measures may warrant anti-TNF therapy, methotrexate, vedolizumab, ustekinumab, short courses of systemic corticosteroids, or surgery on an individual basis.

Decisions on resuming immunosuppressive therapy post-cancer treatment should account for cancer type, recurrence risk, drug-specific oncologic risks, time since treatment completion, and IBD activity, ideally within a multidisciplinary framework. High-risk oncologic patients may require conservative approaches, while low-risk patients may be candidates for earlier reintroduction of advanced IBD therapies.

Evidence from transplant literature suggests delaying immunosuppressant resumption for at least two years post-cancer treatment, extending to five years for cancers with intermediate or high recurrence risk. Most recurrences occur within the first two years: 33% within two years, and 13% beyond five years [92]. Updated ECCO guidelines emphasize case-by-case evaluation by multidisciplinary teams [2].

Practical recommendations include prioritizing cancer control while considering IBD severity and patient quality of life, which may influence adherence to cancer therapy. When cancer is uncontrolled, immunotherapy should be withheld and chemotherapy initiated. If IBD flares during chemotherapy, corticosteroids are first-line therapy, with vedolizumab or ustekinumab as second-line options; accumulating evidence also supports cautious anti-TNF use [94,112]. When cancer is controlled but IBD remains active, corticosteroids are preferred during the first two years, followed by ustekinumab or vedolizumab as second-line therapy. Beyond two years, vedolizumab and ustekinumab are considered safe while awaiting further data on anti-TNF use in this context (see Table 5).

## 5. Discussion

The relationship between IBD, malignancy, and therapy is complex, underscoring the need for individualized, risk-stratified care.

First, although the absolute cancer risk in IBD is low, it rises substantially in patients with long disease duration, extensive colitis, or PSC. Early and tailored colonoscopic surveillance remains critical. Clinicians should ensure structured recall systems, patient education, and coordination between gastroenterology and primary care to improve adherence and outcomes.

Second, treatment selection should balance efficacy with oncologic safety. Thiopurines are best avoided in high-risk or cancer-experienced patients, while anti-TNF agents, vedolizumab, and ustekinumab have favorable long-term safety data. JAK inhibitors and S1P modulators warrant caution and monitoring but should not be categorically excluded.

Third, in patients with prior malignancy, management should be individualized and multidisciplinary rather than guided by rigid restrictions. Emerging data support the safe use of biologics when coordinated with oncology teams to maintain disease control without compromising cancer outcomes.

Fourth, awareness of cancer therapy–IBD interactions is essential. Immune checkpoint inhibitors can trigger IBD flares, highlighting the importance of early gastroenterology involvement in oncology care pathways.

Overall, clinicians should apply vigilant surveillance, thoughtful therapeutic selection, and close interdisciplinary collaboration to minimize cancer burden while sustaining effective IBD control. Ongoing registries such as I-CARE and SAPPHIRE will further refine these strategies and inform best practices.

## 6. Conclusions

Inflammatory bowel diseases (IBDs) are lifelong conditions often requiring immunomodulators and biologic therapies. Because life expectancy is generally unaffected, patients face a baseline risk of cancer development, which may be elevated due to the disease itself (e.g., colorectal cancer) or as a consequence of treatment (e.g., lymphomas and non-melanoma skin cancers associated with thiopurines). Consequently, healthcare providers frequently encounter the challenge of managing IBD therapies in patients with current or past cancer diagnoses. Given the scarcity of controlled trials and limited IBD-specific data, much of the evidence is extrapolated from rheumatology and transplant populations. A multidisciplinary, individualized approach is essential to carefully balance risks and benefits in this complex clinical scenario.

## Figures and Tables

**Table 1 cancers-17-03475-t001:** Cancer with a reported increased risk in IBD: − = not increased risk; + = increased risk; +/− = inconsistent report for increased risk; ++ = major increase of risk.

Cancer Site	Ulcerative Colitis	Crohn’s Disease
Colorectal	++	+
Small bowel	−	++
Cholangiocarcinoma	+	+/−
Upper GI	−	+/−
Lung	−	+/−
Bladder	−	+/−
Leukemia	+/−	−
Non-Hodgkin lymphoma	−	+/−
Melanoma and non-melanoma skin cancer	+	+

**Table 2 cancers-17-03475-t002:** Risk stratification for Colorectal cancer (additional risk factors such as age at diagnosis < 18 years and male gender can also be taken into account).

High Risk	Intermediate Risk	Low/Population Risk
Moderate-severe endo/histo activity PSC (also after transplant) Stricture in the last 5 years Dysplasia in the last 5 years 1st degree relative with CRC	Mild endo/histo activity Extensive disease Post/inflammatory polyps	Proctitis Colitis affecting < 50% of the colon

**Table 3 cancers-17-03475-t003:** Risk of cancers related to therapy for IBD (* = awaiting long-term data).

Medication	Demonstrated Correlation	Dubious Correlation
Thiopurines	Lymphoma, NMSC	Urinary tract, Female genital, Breast
Anti-TNF	Lymphoma (when with thiopurines)	Melanoma, Lymphoma
Vedolizumab	None	
Ustekinumab	None	NMSC
Il-23 inhibitors	None *	None *
JAKi	None	Melanoma, NMSC
S1P modulators	None *	None *

**Table 4 cancers-17-03475-t004:** Cancer recurrence risk according to the type of cancer (can be modified by stage, effectiveness of therapy, tumor grade, lymph node involvement, and presence of specific biomarkers, age, overall health, and lifestyle factors).

High	Intermediate	Low
Glioblastoma	Uterine body	ER-positive breast
Epithelial Ovarian	Prostate	some childhood cancer
Soft tissue sarcoma		HER2-negative breast
Bladder		Thyroid
Pancreas		Kidney (asymptomatic)
Diffuse large B-cell lymphoma		Testicle
Peripheral T-cell lymphoma		Uterine cervix

**Table 5 cancers-17-03475-t005:** Suggestions for management of IBD patients with cancer. (# = Expert consensus; * = Available evidence).

# Medication to Avoid	* Never Use Again
Thiopurines, biologics, and small molecules during active cancer therapy	Thiopurines in HPV or EBV-related cancers
Thiopurines, biologics, and small molecules in the first 2 years after diagnosis	Anti-TNF in case of melanoma
Dual therapy	Thiopurines or anti-TNF after lymphoma

## Data Availability

No new data were created or analyzed in this study. Data sharing is not applicable to this article.

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
