# Peer review of "Management of Inflammatory Bowel Disease with History of Cancer"

_cancers, 2025, doi:10.3390/cancers17213475_

Round 1
Reviewer 1 Report
Comments and Suggestions for Authors
I suggest the following minor revisions to improve clarity and usability:
- Consider repositioning section 3.8 Management, which currently sits under Therapy for IBD and Risk of Cancer, to a more suitable location (perhaps between sections 2 and 4) to improve the logical flow of the manuscript.
- Adding a summary table for section 3 would help readers better visualize the associations between different therapies and cancer risks.
- In Table 4, which discusses management in patients with IBD and cancer, it would be helpful to clarify whether the medications suggested to be avoided are based on available evidence or expert consensus, or if they are mentioned due to lack of sufficient data. This clarification would help avoid confusion and improve the interpretability of the recommendations.
Author Response
Reviewer 1
I suggest the following minor revisions to improve clarity and usability:
- Consider repositioning section 3.8 Management, which currently sits under Therapy for IBD and Risk of Cancer, to a more suitable location (perhaps between sections 2 and 4) to improve the logical flow of the manuscript.
Reply: Many thanks indeed for the suggestion. Indeed each section of “management” is following each of three main pillars of the review, to suggest practical handling in those scenarios: 1) Background risk of cancer; 2) Risk of cancer related to therapy; 3) Situation of previous or new diagnosed cancer. Therefore we believe appropriate the positioning of that para and to clarify we have better explained in the title the meaning of each management paras (i.e. Management of cancer risk related to therapy).
- Adding a summary table for section 3 would help readers better visualize the associations between different therapies and cancer risks.
Reply: Many thanks indeed for the excellent suggestion. Another table (N. 3) has been added to better highlight the cancer risk related to therapy.
- In Table 4, which discusses management in patients with IBD and cancer, it would be helpful to clarify whether the medications suggested to be avoided are based on available evidence or expert consensus, or if they are mentioned due to lack of sufficient data. This clarification would help avoid confusion and improve the interpretability of the recommendations.
Reply: Once again excellent suggestion. This information has been added in the table now numbered as 5 (# = Expert consensus; * = Available evidence)
Reviewer 2 Report
Comments and Suggestions for Authors
This is a well-structured and comprehensive narrative review addressing a highly relevant and clinically challenging topic—the management of inflammatory bowel disease (IBD) in patients with a prior or concurrent history of cancer. The manuscript demonstrates strong command of the literature and integrates recent data from major registries and consensus guidelines, notably ECCO updates and the I-CARE and SAPPHIRE studies. The flow is logical, and the writing is clear, though at times dense. The authors’ expertise is evident, and the review offers meaningful clinical guidance for multidisciplinary decision-making.
The paper is suitable for publication after minor revisions aimed at improving clarity and focus. A few sections could benefit from slight condensation to reduce redundancy (especially in the subsections on thiopurines and anti-TNF therapies), and the discussion could more explicitly summarize practical recommendations for clinical application. No major methodological or conceptual concerns are apparent, and the reference list is up to date and appropriately curated.
Overall, this is a solid and informative review that aligns with the journal’s standards and readership interests. I recommend acceptance after minor editorial adjustments to enhance readability and ensure consistency in terminology and citation formatting.
Author Response
Reviewer 2
This is a well-structured and comprehensive narrative review addressing a highly relevant and clinically challenging topic—the management of inflammatory bowel disease (IBD) in patients with a prior or concurrent history of cancer. The manuscript demonstrates strong command of the literature and integrates recent data from major registries and consensus guidelines, notably ECCO updates and the I-CARE and SAPPHIRE studies. The flow is logical, and the writing is clear, though at times dense. The authors’ expertise is evident, and the review offers meaningful clinical guidance for multidisciplinary decision-making.
The paper is suitable for publication after minor revisions aimed at improving clarity and focus. A few sections could benefit from slight condensation to reduce redundancy (especially in the subsections on thiopurines and anti-TNF therapies), and the discussion could more explicitly summarize practical recommendations for clinical application.
Reply: many thanks indeed for the comment and suggestion. Those paras have been reviewed and modified accordingly.
No major methodological or conceptual concerns are apparent, and the reference list is up to date and appropriately curated.
Overall, this is a solid and informative review that aligns with the journal’s standards and readership interests. I recommend acceptance after minor editorial adjustments to enhance readability and ensure consistency in terminology and citation formatting.
Reply: Many thanks indeed we have double checked terminology and citation formatting.
Reviewer 3 Report
Comments and Suggestions for Authors
This is a well-structured, comprehensive, and clinically relevant review addressing the complex intersection between inflammatory bowel disease (IBD) and cancer management. The paper integrates recent evidence, including updated ECCO guidelines and new registry data, into a coherent narrative. It succeeds in synthesizing data on cancer risk in IBD and providing a framework for management in patients with a prior malignancy, an area with limited randomized evidence but high clinical importance. However, I still have some concerns which need to be addressed before its publication.
Major comments.
- Sections 3.8 and 4.4 related to management overlap conceptually. The reader might struggle to differentiate management of active cancer versus post-cancer IBD.
- For Table 1, it is a good summary but consider harmonizing the “+/-” symbols with descriptive terms (“increased,” “no increase,” “uncertain”). It is not clear to understand “+” and “++”.
- For Table 4, it is a valuable clinical reference. So, I would suggest simplifying presentation by grouping drugs by class (e.g. thiopurines, biologics, small molecules) if possible. Simultaneously, authors should simply provide rationale for avoidance.
- For Discussion and Conclusions sections, the discussion is clearly written, summarizing key learning points. However, some ideas are repeated between the Discussion and Conclusions (e.g., “IBD patients face baseline cancer risk,” “multidisciplinary management is essential”). It would be helpful to merge overlapping content and end with a short “clinical takeaway” paragraph summarizing key recommendations.
- This study included 4 tables. Although the key points can be summarized, adding figures if possible would be helpful for readers.
Minor comments
- For lines 21-22, revise “health-care givers treating IBD can frequently be faced with a previous or incident cancer history” to “clinicians managing IBD frequently encounter patients with a prior or newly diagnosed cancer” or similar to make it flower.
- For line 21, “long disease life span” sounds a bit weird. Consider rephrasing.
Author Response
Reviewer 3
This is a well-structured, comprehensive, and clinically relevant review addressing the complex intersection between inflammatory bowel disease (IBD) and cancer management. The paper integrates recent evidence, including updated ECCO guidelines and new registry data, into a coherent narrative. It succeeds in synthesizing data on cancer risk in IBD and providing a framework for management in patients with a prior malignancy, an area with limited randomized evidence but high clinical importance. However, I still have some concerns which need to be addressed before its publication.
Reply: many thanks indeed for your kind comment
Major comments.
- Sections 3.8 and 4.4 related to management overlap conceptually. The reader might struggle to differentiate management of active cancer versus post-cancer IBD. Reply: many thanks indeed for this comment. Indeed, it could be a bit confusing. In the revised version, the title for para 3.8 and 4.8 has been expanded to clarify the context, such as: Management of cancer risk related to therapy and Management of therapy in patients with IBD and cancer
- For Table 1, it is a good summary but consider harmonizing the “+/-” symbols with descriptive terms (“increased,” “no increase,” “uncertain”). It is not clear to understand “+” and “++”.
Reply: many thanks indeed for the suggestion. We have clarified the meaning of symbols in the table legenda. . (- = not increased risk; + = increased risk; +/- = inconsistent report for increased risk; ++ = major increase of risk)
- For Table 4, it is a valuable clinical reference. So, I would suggest simplifying presentation by grouping drugs by class (e.g. thiopurines, biologics, small molecules) if possible. Simultaneously, authors should simply provide rationale for avoidance. Reply: many thanks indeed for the valuable suggestion. Table 4 is now number as Table 5 (a new table 3 has been added). We have provided in the legenda the rationale for avoidance (# = Expert consensus; * = Available evidence). We have also avoided the terminology immunomodulation and mentioned the different drugs.
- For Discussion and Conclusions sections, the discussion is clearly written, summarizing key learning points. However, some ideas are repeated between the Discussion and Conclusions (e.g., “IBD patients face baseline cancer risk,” “multidisciplinary management is essential”). It would be helpful to merge overlapping content and end with a short “clinical takeaway” paragraph summarizing key recommendations.
Reply: Thanks to raise this point; we had a similar comment from the reviewer # 2. The para of discussion has been completely rewritten with more clear takeaway clinical points.
- This study included 4 tables. Although the key points can be summarized, adding figures if possible would be helpful for readers. Reply: Thanks for the suggestion. We have added a further table to better clarify the contents. Not sure how the key-points can be translated in a figure. I’m looking forward for a suggestion if any form the Editorial Staff.
Minor comments
- For lines 21-22, revise “health-care givers treating IBD can frequently be faced with a previous or incident cancer history” to “clinicians managing IBD frequently encounter patients with a prior or newly diagnosed cancer” or similar to make it flower.
Reply: Thanks for the suggestion we have modified the sentence as suggested.
- For line 21, “long disease life span” sounds a bit weird. Consider rephrasing.
Reply: Thanks, we have rephrased as follow: Because of the chronic course of the disease
Round 2
Reviewer 3 Report
Comments and Suggestions for Authors
Concerns were addressed. Congratulations!